

# Identification of cytokine-induced cell communications by pan-cancer meta-analysis

Yining Liu[1], Min Zhao[2] and Hong Qu[3]

[1] The School of Public Health, Institute for Chemical Carcinogenesis, Guangzhou Medical University, Guangzhou, China
[2] School of Science and Engineering, University of the Sunshine Coast, Maroochydore DC, Australia
[3] Center for Bioinformatics, State Key Laboratory of Protein and Plant Gene Research, College of Life Sciences, Peking University, Beijing, China

## ABSTRACT

Cancer immune responses are complex cellular processes in which cytokine–receptor interactions play central roles in cancer development and response to therapy; dysregulated cytokine–receptor communication may lead to pathological processes, including cancer, autoimmune diseases, and cytokine storm; however, our knowledge regarding cytokine-mediated cell–cell communication (CCI) in different cancers remains limited. The present study presents a single-cell and pan-cancer-level transcriptomics integration of 41,900 cells across 25 cancer types. We developed a single-cell method to actively express 62 cytokine–receptor pairs to reveal stable cytokine-mediated cell communications involving 84 cytokines and receptors. The correlation between the sample-based CCI profile and the interactome analysis indicates multiple cytokine–receptor modules including *TGFB1*, *IL16ST*, *IL15*, and the *PDGF* family. Some isolated cytokine interactions, such as *FN1–IL17RC*, displayed diverse functions within over ten single-cell transcriptomics datasets. Further functional enrichment analysis revealed that the constructed cytokine–receptor interaction map is associated with the positive regulation of multiple immune response pathways. Using public TCGA pan-cancer mutational data, co-mutational analysis of the cytokines and receptors provided significant co-occurrence features, implying the existence of cooperative mechanisms. Analysis of 10,967 samples from 32 TCGA cancer types revealed that the 84 cytokine and receptor genes are significantly associated with clinical survival time. Interestingly, the tumor samples with mutations in any of the 84 cytokines and receptors have a substantially higher mutational burden, offering insights into antitumor immune regulation and response. Clinical cancer stage information revealed that tumor samples with mutations in any of the 84 cytokines and receptors stratify into earlier tumor stages, with unique cellular compositions and clinical outcomes. This study provides a comprehensive cytokine–receptor atlas of the cellular architecture in multiple cancers at the single-cell level.

Corresponding authors
Min Zhao, mzhao@usc.edu.au
Hong Qu, quh@mail.cbi.pku.edu.cn

## INTRODUCTION

Cytokines play essential roles in the immune response by mediating the activation of proliferation and signaling in the innate and adaptive immune systems (*Cameron & Kelvin, 2000–2013*). Recent studies have shown that cytokines are associated with cancer development by aiding the formation of the tumor microenvironment (*Kumar et al., 2022*; *Lee & Margolin, 2011*; *Lin, Sharma & John, 2014*). Cytokines exert their effects by binding to specific receptors on target cells, thereby activating intracellular signaling pathways; however, cell–cell interactions are dynamic and transient, and only a few technologies are currently capable of capturing these interactions (*Zhang et al., 2022*). As major players in immune regulation and inflammation, interleukins, interferons, tumor necrosis factors, colony-stimulating factors, growth factors, and chemokines can be used to treat cancer (*Habanjar et al., 2023*). To date, more than 20 different cytokine medications have been approved and commercialized worldwide for the prevention and treatment of various diseases (*Habanjar et al., 2023*; *Propper & Balkwill, 2022*; *Wolfarth et al., 2022*).

Interleukins are produced by lymphocytes, macrophages, and other cells, and play a vital role in the control of immunity by modulating T-cell and B-cell activation, proliferation, and differentiation (*Martinez-Perez et al., 2021*). Interferons exert an anticancer function; for instance, IFNα induces tumor cell death *via* the Fas pathway and is successfully used as a tumor immunotherapy (*Kimura et al., 2003*). In general, tumor necrosis factors (TNFs) suppress P53 function and facilitate cell apoptosis (*Joo et al., 2023*). In addition, colony-stimulating proteins, such as CSF-1 and their receptors, increase tumor cell proliferation and survival in an autocrine or paracrine manner (*Achkova & Maher, 2016*). Moreover, growth factors are a type of cytokine that regulates cell growth and differentiation. VEGF (vascular endothelial growth factor) is one of the most significant factors for tumor angiogenesis since it provides the necessary oxygen and nutrients while also mediating tumor proliferation activity (*You et al., 2023*). Furthermore, chemokines can stimulate target cells to migrate in a specific direction (*Kim et al., 2011*).

Specific proteins on the cell surface mediate transient cell–cell interactions (CCIs), which are short-lived contacts between cells (*Gutwillig et al., 2022*). For cytokine-related transient CCIs, cells synthesize and secrete cytokines to readily perform their biological roles only in response to tissue damage and infection (*Srivastava & Rasool, 2022*). The vast majority of cytokine receptors are membrane-spanning proteins containing extracellular, transmembrane, and cytoplasmic domains. According to the three-dimensional structure of cytokine–receptor complexes, the binding of cytokines to the receptor occurs through electrostatic and van der Waals interactions, and no stable chemical bonds are formed. By understanding the molecular mechanisms that mediate these cytokine-based stable CCIs in multiple cells, we may gain deeper knowledge regarding the manner by which immune cells communicate and how these interactions contribute to the development of cancer.

Single-cell RNA sequencing (scRNA-seq), a new method for high-throughput mRNA sequencing at the single-cell level, can effectively determine cell–cell communications (CCIs) that cannot be deciphered from tissue samples. Moreover, scRNA-seq solves the issue of transcriptome heterogeneity within cell populations that is masked by conventional

RNA-seq, allowing new rare cell types to be discovered (*Liu & Zhao, 2021*). Despite many studies linking various types of cytokines to carcinogenesis, no comprehensive evaluation of cytokine-mediated CCIs in cancers has been performed to date. Here, we conducted the first systematic exploration of cytokines in 41,900 cells across 25 cancer types. The constructed stable cytokine–receptor CCI network provides a heterogenetic view of tumorigenesis and the tumor immune microenvironment, providing a basis for cytokine-based drug research and clinical application, which may more precisely direct future treatments.

## MATERIALS AND METHODS

### Collecting the single-cell transcriptomes across multiple cancer types

To explore the heterogeneity of cell–cell communications, we conducted a survey of single-cell transcriptomes at the pan-cancer level. In the present study, we adopted a scRNA-seq database CancerSEA, which contains 41,900 single-cell transcriptomes from 25 cancer types (*Yuan et al., 2019*). In total, CancerSEA has collected 72 datasets by focusing on scRNA-seq data. In practice, we downloaded the processed transcripts per kilobase million (TPM) value after quality control and normalization of each dataset on 20th May 2021. Subsequently, log2 transformation was applied to all the TPM expression values. In addition to gene expression, CancerSEA also depicts 14 functional states related to cancer development: angiogenesis, apoptosis, cancer stemness, cell cycle, cell proliferation, differentiation, DNA damage and repair, epithelial–mesenchymal transition, hypoxia, inflammation, invasion, metastasis, and quiescence at the single-cell level across 25 cancer types. The general idea was to collect a list of signature genes associated with certain functional states. The activities of all the functional states of a cell were then calculated using the Gene Set Variation Analysis (GSVA) package in R (*Hanzelmann, Castelo & Guinney, 2013*). In a word, these single-cell transcriptomes and cell status data serve as a platform to explore the cell–cell communication from a cytokine–receptor perspective.

### Curating the list of cytokines and receptors

To explore the cytokine–receptor interactome, we curated a list of cytokines and their classification from an online resource (*Cameron & Kelvin, 2000–2013*). The receptors of these cytokines were integrated from OmniPath (*Ceccarelli et al., 2020*). We collected a total of 328 unique cytokine–receptor interaction pairs involving a total of 122 cytokines and 126 receptors. It is important to note that multiple cytokines may interact with multiple receptors and vice versa.

### Determining active gene expression and building the initial cell–cell communication networks

Since droplet-based microfluidics measures gene expression in the transcriptome of a single cell, a lowly expressed transcript is frequently undetected and therefore assigned a value of zero (*Linderman et al., 2022*). Accordingly, we established a mechanism for filtering active genes at the level of a single cell. In each dataset, the log2 transformed TPM value of a gene must be 1 or greater.

It was necessary to extract actively expressed cytokine and receptor genes in order to construct the fundamental framework of the cell–cell communication network in each

dataset. We then joined cell pairs together if one of the cells actively expressed a cytokine and the other actively expressed a receptor specific to that cytokine. Subsequently, all possible CCIs for each dataset could be connected using this method. Only one of the 72 datasets downloaded from CancerSEA lacked both actively expressed cytokines and receptors and was therefore filtered out.

In summary, there are a total of 71 datasets based on actively expressed cytokine–receptor pairs, with a combined total of 86,923,583 cell–cell communications in 24 different types of cancer. These preliminary CCIs include a total of 266 distinct active interacting pairs involving 112 receptors and 104 cytokines.

### Defining stable cell–cell communications

To prevent the collection of transient CCIs, we focused on stable cell–cell communications. In brief, we calculated the stable cell communication ratio based on the active expression of cytokines and receptors in a dataset using Eq. (1):

$$s = m/n^2 \qquad (1)$$

where $s$ is the stable expression ratio indicating the stability of the CCIs initiated by a specific cytokine–receptor in a dataset; $m$ is the number of CCIs based on actively expressed cytokine–receptor patterns; $n$ depicts the total cell number in a specific dataset; and $n^2$ indicates all theoretical CCIs in the given dataset. By focusing on an $s$ value over 0.5, we defined a cytokine–receptor CCI as stable and applied the criteria to all 71 datasets to collect all the stable CCIs and corresponding cytokine–receptor pairs.

### Functional analysis and clinical application

For the 84 stably expressed cytokines and receptors, the functional features were further explored based on the publicly available gene ontology and pathway annotations using the R package GOSemSim (*Yu et al., 2010*). Briefly, we installed GOSemSim in R (version 4.3; *R Core Team, 2023*) (2022.07.1 Build 554) and performed enrichment analysis (*Boyle et al., 2004*); the $P$ value was determined using the hypergeometric distribution. Only those GO terms with two or more genes from the list of 84 cytokines or receptors were considered when attempting to determine the significance of the GO terms. The estimated significance level was adjusted for multiple hypothesis testing using the FDR control. A bubble plot was generated, in which the colors represent adjusted $P$ values and bubble size represents gene counts.

To explore the mutational and clinical features, we mapped all 84 genes to the public cancer genomics data using TCGAbiolinks (*Colaprico et al., 2016*). In total, the TCGA pan-cancer database contains 10,967 samples across 32 cancer types. The clinical data including survival time, tumor burden, and cancer stage information were extracted from the samples with genetic variations of the 84 cytokines and receptors. Kaplan–Meier survival curves were generated based on survival time. For multi-stage information, a stacked bar chart was generated to present the distribution of mutated samples.

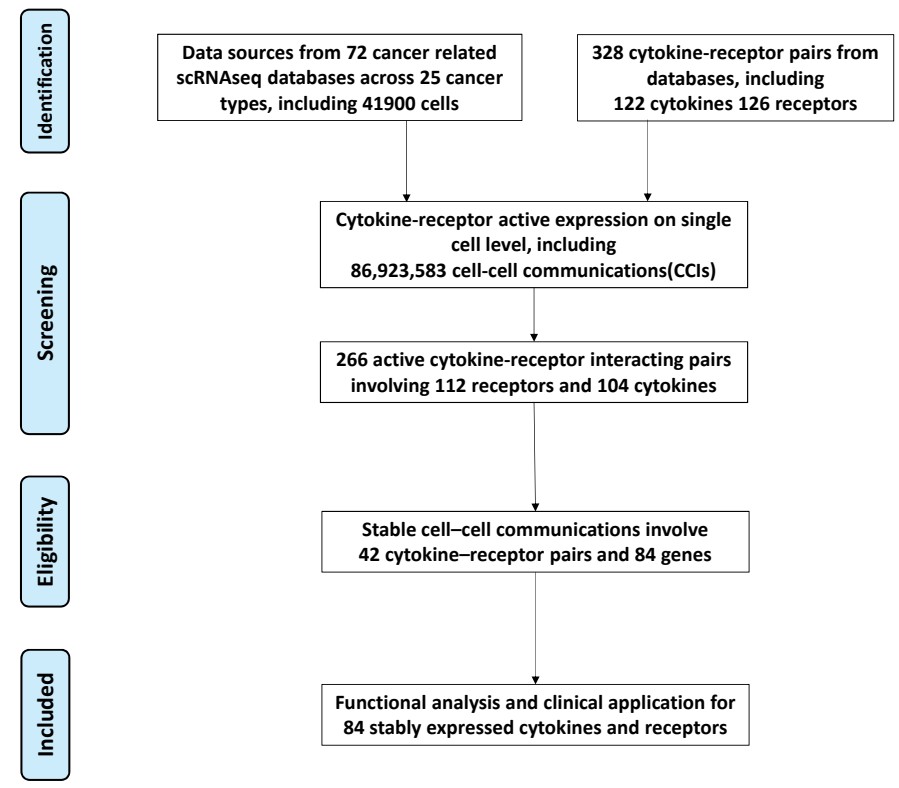

**Figure 1** PRISMA flow diagram for the construction of stable cell communications induced by cytokine–receptor interactions.

## RESULTS

### Computational workflow used to explore the cytokine–receptor map on a single-cell level in 41,900 cancer cells

Figure 1 shows the PRISMA flow diagram for the construction of stable cell communications induced by cytokine–receptor interactions. To explore the CCIs related to cancer immunology, we established a computational workflow by focusing on cytokines and their receptors (Fig. 2). The pipeline started from characterization of the expression levels of cytokines and their receptors at the single-cell level. In brief, we collected a total of 328 interaction pairs with 122 cytokines and 126 receptors based on reviews and public databases. All expression data were normalized within the dataset and used to define actively expressed cytokines and receptors (see Methods). Accordingly, 266 unique active interacting pairs between 104 cytokines and 112 receptors were extracted. Subsequently, the CCIs were constructed based on the active expression of cytokines in one cell and the active expression of known receptors in another cell from multiple cancer scRNA-seq datasets.

   To prevent the collection of transient CCIs (*Liu & Zhao, 2021*), we focused on stable cell communications initiated by cytokine–receptor interaction pairs, which were supported by over 50% of all the CCIs in a dataset. For instance, the cytokine–receptor pair *TGFB1–SDC2*

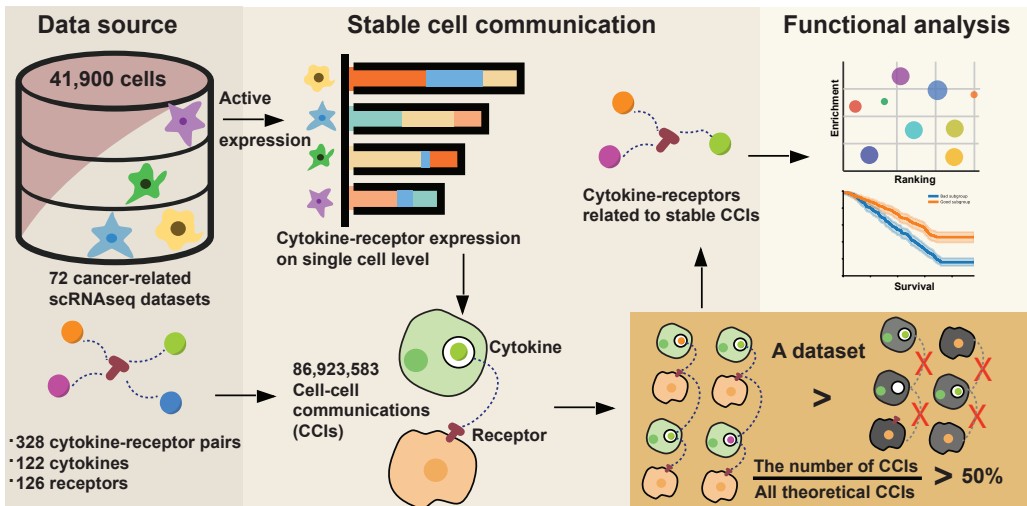

**Figure 2** The computational workflow for the construction of stable cell communications induced by cytokine–receptor interactions.

initiated the most CCIs (18,054,208 CCIs) in an alveolar rhabdomyosarcoma scRNA-seq dataset (*Chen et al., 2018*). In this dataset, the expression patterns of 6,875 single cells were characterized; therefore, in theory, there were a total of 47,265,652 (6,875 × 6,875) CCIs. By checking the stable CCI ratio (18,054,208 ÷ 47,265,652), *TGFB1–SDC2* is only involved in approximately 38% of all conceivable CCIs. Based on our stable CCI criteria, we filtered this cytokine–receptor out in the final stable interaction map. Accordingly, the outcome was 239 detectable and stable cytokine–receptor interaction pairs across 35 datasets, among which 62 unique interaction pairs were detected consisting of 42 cytokines and 42 receptors.

## The stable cytokine–receptor interaction map in cancers

By executing the computational workflow, we focused on all the unique cytokine–receptor pairs to generate a cytokine–receptor interaction map with substantial CCI traffic in cancers (Fig. 3A). In total, these stable CCIs involve 42 cytokine–receptor pairs and 84 genes. To evaluate whether the 42 cytokines and 42 receptors initiate large-scale cell communication, their molecular relationships were investigated. It should be noted that this interaction map is presented at the molecular level; however, each link on the map could represent hundreds of cell–cell communications. The hub nodes of a network are frequently used as common links to handle information transmission over a short distance. In our map, 17 genes are completely connected to *TGFB1* as the hub node; therefore, it is expected that *TGFB1* has pleiotropic effects on inflammation, cell growth, and differentiation (*Tewari et al., 2022*). Regarding the hundreds and thousands of CCIs behind these 17 genes, the results suggest that targeting *TGFB1* may be an effective cancer prevention treatment to block communication between tumor cells.

In addition, a further 11 cytokines and receptors are connected *via IL6ST* as the node, which has also been reported as a prognostic biomarker in breast cancer (*Martinez-Perez*

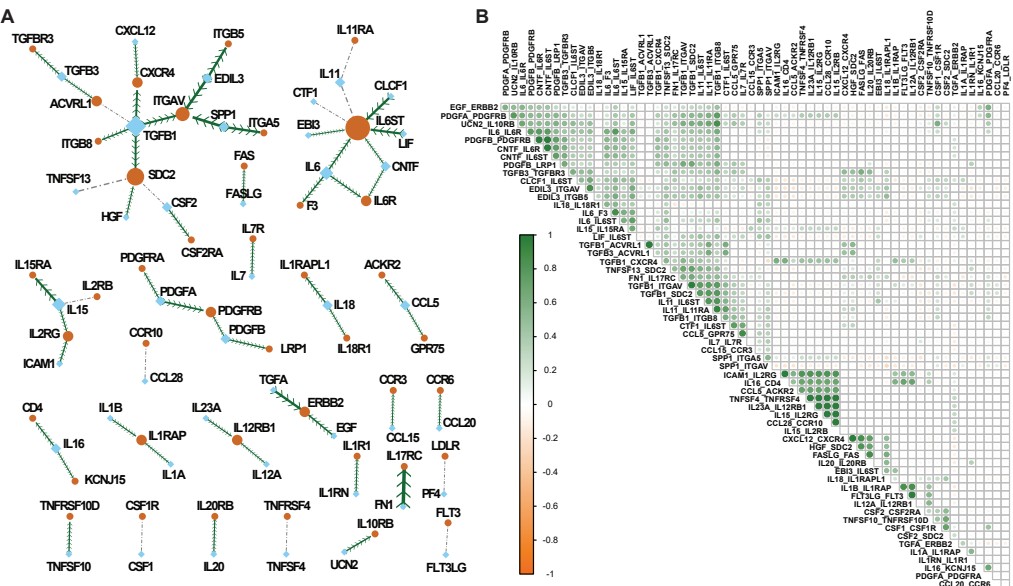

**Figure 3  The stable cytokine–receptor map for cancer cell communications.** (A) The cytokine–receptor interaction network summarized from the stable cell communications based on 41,900 samples combined from 72 studies. The size of each node is correlated with the number of connections. The orange nodes are cytokines and the blue nodes are receptors. The edge width is correlated with the number of associated datasets. The edges with arrows are significantly co-occurring pairs based on TCGA pan-cancer mutational analysis. (B) The correlation plot for the 42 cytokine–receptor pairs based on the number of cell communications in 72 datasets.

*et al., 2021*). With emerging roles in infection, chronic inflammation, autoimmunity, and cancer, members of the IL-6 family may be ideal therapeutic targets for the manipulation of disease states (*Jones & Jenkins, 2018*). Some other modules are also centred around *IL15* and the platelet-derived growth factor (PDGF) family. As one of the most common $\gamma$-chain cytokines, IL15 has great potential as an cancer immunotherapy (*Wolfarth et al., 2022*). Moreover, since PDGFs and their receptors are expressed in a variety of tumors, these proteins frequently play crucial roles in cancer proliferation, metastasis, invasion, and angiogenesis (*Zou et al., 2022*).

The remaining cytokines are mostly connected to their specific receptors. Despite the *FN1–IL17RC* pair not being connected to any other cytokines or receptors, it was detected in 16 datasets, indicating that this interaction may have a broader function in multiple cancers. For example, a high expression level of fibronectin 1 (*FN1*) is associated with poor prognosis in gastric cancer (*Sun et al., 2020*). Our data suggest that *FN1–IL17RC* may play additional roles in acute lymphoblastic leukemia, bronchoalveolar carcinoma, breast cancer, cervical cancer, glioblastoma, hepatocellular carcinoma, melanoma, non-small cell lung cancer, and prostate cancer.

A recent systematic study revealed that co-mutations with a higher prognostic value have a higher potential impact, implying a cooperative mechanism for tumorigenesis (*Jiang et al., 2022*). By leveraging large-scale cancer genomics data, we also conducted co-mutational

analysis based on 10,967 genes from 32 TCGA cancer types. Among the 62 connections, two genes in 50 pairs were significantly co-mutated. Further, based on the number of CCIs, we also conducted correlation analysis of all the 62 cytokine–receptor pairs (Fig. 3B).

## Cancer type heterogeneity

We further investigated the CCIs from the top cytokine–receptor modules based on the cancer types with which they are associated, allowing us to provide an overview of the heterogeneity of cytokine-initiated CCIs in a number of different cancers. Based on the 61 different interacting cytokine–receptor pairs, a total of 2,266,386 stable CCIs were generated. We analyzed the similarities and differences between the CCIs that were induced by various cytokines and receptors by mapping these CCIs to various cancer types.

Since *TGFB1* and *IL6ST* are the two modules with the highest degree of molecular connectivity, we compared the cancer types with which each of these modules is associated (Fig. 4A). In general, the distribution of cancer types within these two modules is comparable to that of all 61 cytokine–receptor pairs. This holds true for both aforementioned modules. Cancers of the lung and breast, both of which have a greater number of cells and datasets, were found to have a greater number of cytokine–receptor pairs. It is important to note that *TGFB1*- and *IL6ST*-related CCIs were not found in prostate or colorectal cancers included in a single dataset. In summary, these results suggest that the *TGFB1* and *IL6ST* modules may have broader effects in cell communications in multiple cancer types.

The majority of these 61 cytokine–receptor pairs can be found in multiple types of cancer. In total, there are 36 pairs detected in two or more cancer types. The remaining 25 cytokine–receptor pairs are unique to a particular form of cancer and cannot be found elsewhere. Three of the 25 pairs are distinct to a single cancer type in two separate datasets. For example, in two separate datasets pertaining to breast cancer, IL1RN–IL1R1 was responsible for 3,779 and 1,445 new cases, respectively. Dysregulation of the IL-1 proinflammatory cascade has been linked to cancer initiation, progression, and invasiveness; however, its clinical significance in the treatment of cancer remains to be determined (*Litmanovich, Khazim & Cohen, 2018*). In breast cancer, upregulation of the IL-1 receptor is associated with anti-estrogen-resistant cancer stem cells and novel IL-1 antagonists are being developed as a treatment for a variety of cancers, which have shown good safety profiles in the short term; nevertheless, long-term monitoring of adverse events are required to confirm these findings. In most cancer-related clinical trials, inhibition of the IL-1 system has improved symptoms, particularly following the use of novel drugs as adjuvant treatments with chemotherapy (*Litmanovich, Khazim & Cohen, 2018*); however, there exist no dependable data regarding survival improvement or antitumor activity, which may be of greater importance.

In addition, two cytokine–receptor pairs (IL16–CD4 and ICAM1–IL2RG) are uniquely detected in two leukemia datasets. Positioned at the center of the SN-Treg signaling network, IL16 can attract CD4+ T cells and inhibit CD3-mediated lymphocyte activation and proliferation (*Zhang & Xu, 2002*), and recombinant human IL16 has shown inhibitory effects on the growth of human T-cell leukemia Jurkat cells (*Zhang & Xu, 2002*). Moreover,

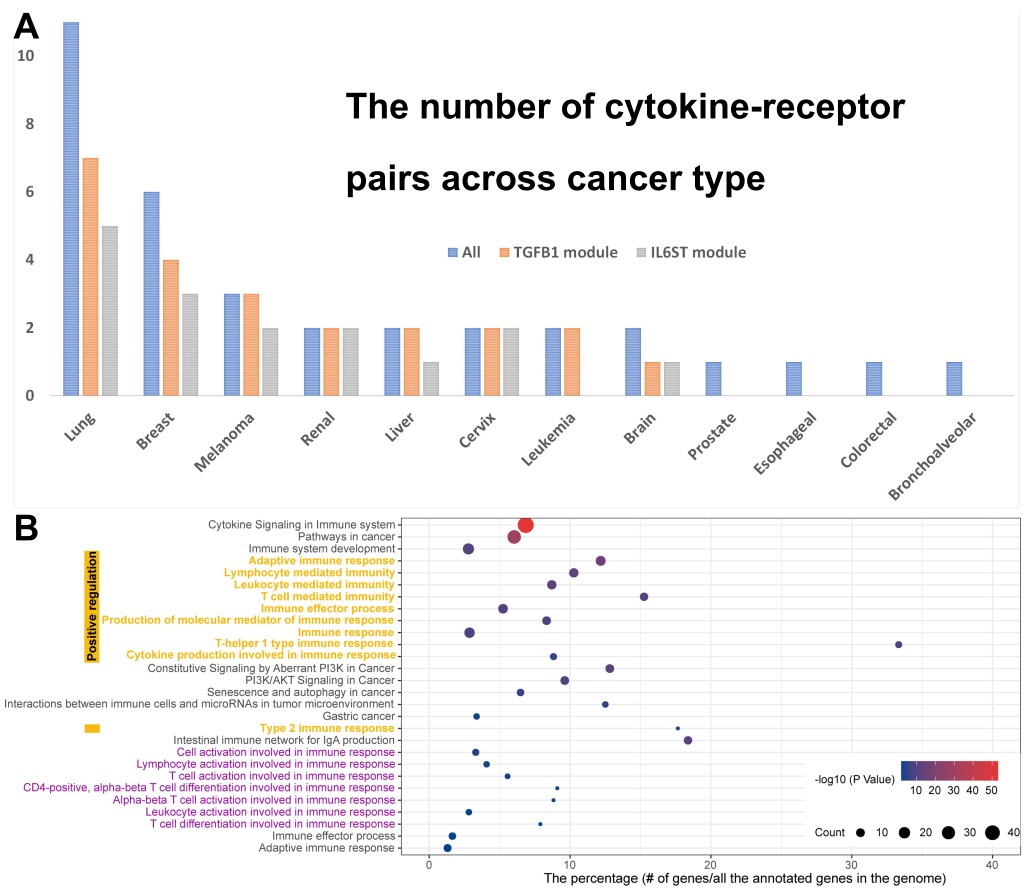

**Figure 4** **The cancer type distribution and functional enrichment for the 84 cytokines and receptors initiating stable cell communications in multiple cancers.** (A) The cancer type distribution. (B) The enriched functional terms related to cancer and immune response. The *x*-axis represents the top enriched functional terms and the *y*-axis represents the percentage of genes out of the total annotated genes in the human genome. The color and size of the nodes are presented at the bottom of the figure.

ICAM1 has been reported to be upregulated in multiple myeloma cells in comparison with normal cells. An anti-ICAM1 antibody–drug conjugate has displayed potent anti-myeloma cytotoxicity *in vitro* and *in vivo* (*Sherbenou et al., 2020*); therefore, this anti-ICAM1 antibody–drug conjugate should be further studied for toxicity and if proven safe, tested for clinical efficacy in patients with relapsed or refractory multiple myeloma.

## Positive regulation of the immune response

To acquire a functional overview of our constructed cytokine–receptor map, we carried out functional enrichment analysis with an emphasis on the pathways involved in the immune response and cancer development (Fig. 4B, Table S1). As expected, a significant association was found between the 84 cytokines and receptors and pathways involved in cancer (corrected $P$-value = $10^{-53}$). Interestingly, more genes are specific to gastric cancer (corrected $P$-value = $10^{-3}$), which was not included in our cancer type distribution. Additionally, there is a pathway connecting cancer and immunology, which
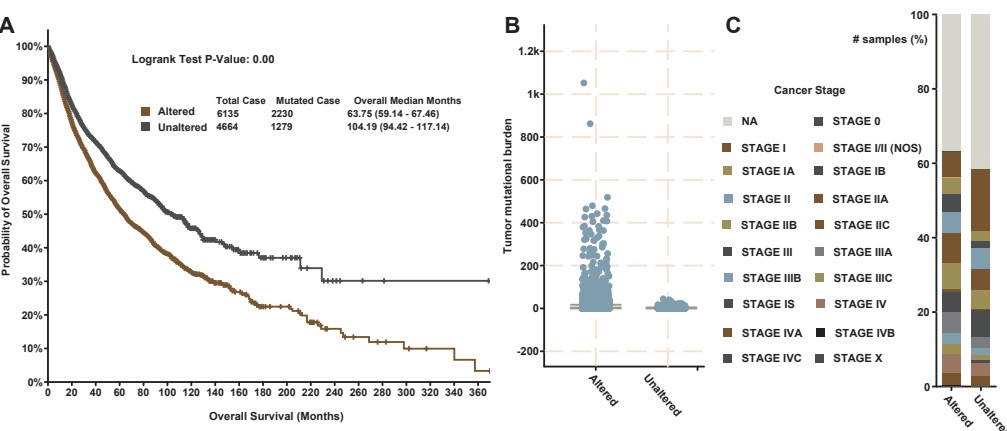

**Figure 5  The clinical features of the 84 cytokines and receptors initiating stable cell communications.**
(A) Survival analysis of the 84 cytokines and receptors in the 10,967 tumor samples from the TCGA pan-cancer cohort. (B) The mutational burden of samples with mutations in the 84 cytokines and receptors (altered group) and those with no mutations (unaltered group). (C) The cancer stage of samples with mutations in the 84 cytokines and receptors (altered group) and those with no mutations (unaltered group).

focuses on interactions that occur between immune cells and microRNAs in the tumor microenvironment (corrected $P$-value $= 10^{-5}$).

Unsurprisingly, a significant number of immune response pathways are enriched among the 84 genes, since one of the primary functions of cytokines is to regulate the immune response. For example, we uncovered 11 positive regulatory roles including the adaptive immune response (corrected $P$-value $= 10^{-16}$), lymphocyte-mediated immunity (corrected $P$-value $= 10^{-13}$), leukocyte-mediated immunity (corrected $P$-value $= 10^{-12}$), and T cell-mediated immunity (corrected $P$-value $= 10^{-11}$). Taken together, these findings provide evidence that our cytokine–receptor interaction map contains a number of pro-immunity factors that have the potential to be utilized in the modulation of cancer immunology. Numerous cytokines have been validated for use in the treatment of cancer, some of which include IL-2 for the treatment of metastatic melanoma and renal cell carcinoma, in addition to IFN as an adjuvant therapy for stage III melanoma (*Lee & Margolin, 2011*).

## Clinical application of the cytokine–receptor interaction map

To further investigate the potential clinical application of our cytokine–receptor interaction map, we carried out a survival analysis based on 10,967 samples taken from TCGA pan-cancer datasets covering 32 different cancer types. This allowed us to examine the potential clinical application of the cytokine–receptor pairs in our stable CCI network. A significant relationship exists between these 84 genes and patient survival (Fig. 5A, Log rank test $P$-value $= 0$). More interestingly, the $P$-values for the 42 cytokines and 42 receptors are both higher than the $P$-value for their combination, confirming that co-mutated cytokine–receptor pairs have higher prognostic value.

Based on genetic mutations in the 84 genes, we were able to classify the samples into two groups. Interestingly, the group with genetic changes has a significantly higher mutational
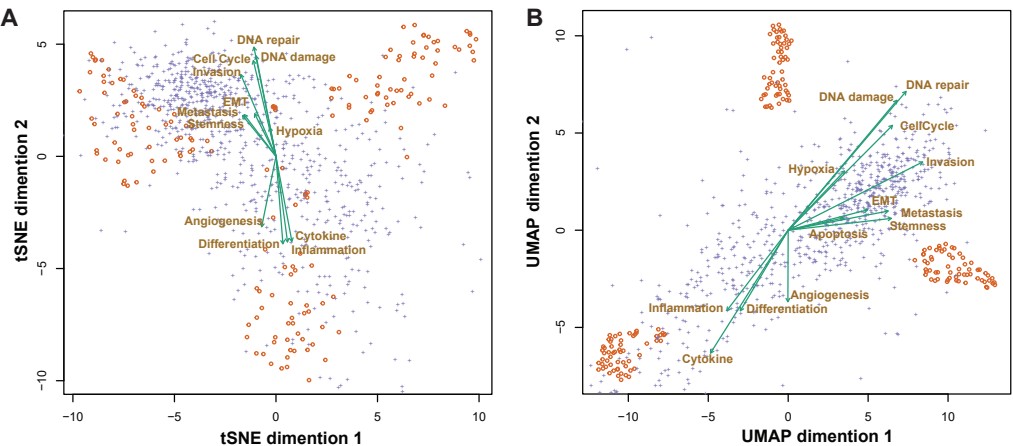

**Figure 6** The t-SNE (A) and UMAP (B) biplots demonstrating the connection between intratumor heterogeneity and cell status in 430 glioblastoma cells. The orange circles represent cells; the purple " + " represents genes with the highest percentage of variants; and green vectors represent the five cell states.

burden (Fig. 5B), which is defined by the number of genetic changes in a cancer cell. In general, more genetic changes may result in the presentation of a greater number of antigens at the cell surface, increasing the chance of an immune response being initiated. This also explains why a high mutational burden in less common solid tumors is often associated with poorer patient survival (*Shao et al., 2020*). On the contrary, those samples with genetic mutations in the 84 cytokines and receptors are more likely to be in the earlier stages, such as stage I (Fig. 5C), indicating that the distribution of mutational burden may correlate with pathological/histological subtype and stage (*Qiu et al., 2020*). In summary, we reveal that cytokines and their associated receptors are more likely to be mutated in late stage cancer, generating a higher mutational burden. Accordingly, these highly mutated cancer cells are more likely to be recognized by immune cells and promote the immune response. Our data provide a new perspective on cytokine–mediated immune regulation.

## Correlation among different cell states

Performing multiple gene-based signature expression analyses allowed us to characterize the relative expression strength in different pathways and identify the cooperative and antagonistic effects between pathways. For illustration purposes, we took a dataset containing glioblastoma samples and evaluated the cytokine and receptor status of each individual cell sample (GSE57872) (*Patel et al., 2014*). In the single-cell transcriptome, thousands of gene expression data from hundreds of cells are frequently present; therefore, it is impossible to observe and compare gene expression in all cells. The simplest method is to reduce the data dimensions. t-SNE and UMAP are the two most efficient techniques for reconstructing data distribution in a lower-dimensional space while maintaining its structure; thus, all genes (purple plus), cells (orange circle), and pathways (teal vector) could be mapped in two dimensions (Fig. 6).

In both the t-SNE and UMAP charts, the cytokine vector follows the same direction as inflammation and differentiation (Fig. 6), and DNA damage and repair are inversely proportional to the cytokine vector. Consequently, the cytokine may have a significant effect on the degree of inflammation and differentiation of glioblastoma cells.

In summary, our pathway-level analysis may provide a novel overview of the relationship among the processed cancer markers, also highlighting immune regulation mediated by cytokines. For example, it has been reported that inflammatory cytokines can induce DNA damage and inhibit DNA repair in cholangiocarcinoma cells *via* a nitric oxide-dependent mechanism (*Zhang & Xu, 2002*). Notably, cytokine signaling diversity within different cancer cell populations is often obscured by conventional RNA-seq in tissue samples, and only scRNA-seq can effectively solve the problem of cytokine-mediated cellular heterogeneity. Thus, our computational approach facilitates the discovery of new, rare cell communications and the acquisition of fresh understanding.

## DISCUSSION

In the present study, we performed the largest pan-cancer meta-analysis of cytokine–receptor interactions at the single-cell level. Based on millions of putative cell communications, we prioritized the most essential cytokine–receptor interactions and constructed a cytokine–receptor interaction map at the single-cell level. By focusing on immunological functions, we found that a number of critical interleukins and growth factors bridge various receptors in multiple cancers. In addition, these cytokine-mediated biological processes are significantly associated with positive regulation of the immune response in different immune cells; therefore, these inflammatory cytokines and counter-regulatory substances are ideal candidates for cancer immunotherapy.

Traditional transcriptome sequencing (bulk RNA-seq) provides the average expression levels of genes in a population of cells but overlooks inter-cell differences; therefore, the detection of cell–cell communications (CCIs) is challenging. On the contrary, scRNA-seq dissociates and sequences tissue at the single-cell level, and it is feasible to accurately determine the expression status of ligands and receptors involved in CCIs. However, in the current single-cell RNA sequencing platforms, such as 10X Genomics, droplet-based microfluidics are frequently used to measure gene expression (*Qiu, 2020*). In theory, droplet technology necessitates the amplification of extremely minute amounts of mRNA, resulting in a phenomenon known as "dropout", in which an expressed transcript is not detected and is therefore assigned a value of zero. In the present study, we focused on the actively expressed cytokines and their receptors.

In cancer cells, transient CCIs play multiple roles in cell signaling, development, and immunity. Each type of transient CCI is mediated by a distinct group of proteins. Immune cells detect and respond to foreign invaders *via* transient CCIs. T cells, for instance, utilize adhesion molecules to bind to antigen-presenting cells, allowing them to receive signals that aid in their anti-infection efforts. Typically, the extracellular N-terminal domain is the site of cytokine recognition. The extracellular 6th and 7th transmembrane helical portions of the receptor have a relatively large degree of freedom to move, as

evidenced by their three-dimensional structure (*Urvas & Kellenberger, 2023*). By oscillating these two helical portions, a structural domain for cytokine binding is formed, thereby activating the receptor. Consequently, it is easy to separate cytokines from their receptors under conditions of fluctuating temperature, pH, and acidity. Based on the preceding explanation, we focused on the cytokine-induced CCIs detected in more than half of the theoretical CCIs in a dataset, thus excluding the vast majority of transient cases. There may be a more effective way to optimize the cutoff for identifying stable CCIs. We believe that methods based on large-scale deep learning have the potential to solve this issue. As a result of limited computational resources, our study arbitrarily selected 50% of the theoretical CCIs in a dataset as the cutoff.

There are existing methods for the construction of CCIs; for example, CellChat is able to deduce the signaling communications that occur between different cell groups according to the genes that are differently expressed in each group (*Jin et al., 2021*). As a result, the output of CellChat is the interaction between cell group 1 and cell group 2, indicating that CellChat is unable to establish cell communication at the single-cell level. In addition, CellChat uses grouping criteria based on cell types, as opposed to tissue origins, to organise cells. It automatically groups cells based on their annotation types. This indicates that CellChat identifies and analyses cell-cell communication patterns and interactions among cells that share similar cell type annotations, irrespective of their tissue of origins. In the present study, we investigated the CCI at the level of a single cell rather than a cell group. In lieu of differentially expressed genes, we proposed a general method based on expression data.

By collecting known cytokine–receptor interaction information, we linked the cells and predicted the stable CCIs, finally identifying 84 cytokines and receptors related to stable CCIs in multiple cancers. Some publications such as CytokineLink, a map of cytokine communication in inflammatory and infectious diseases (*Olbei et al., 2021*), also construct cytokine networks specific to inflammatory bowel disease and COVID-19. Although the tool provides a list of CCIs based on cytokines and receptors, it is extremely broad at the tissue level: source adipose tissue > CCL7 > CCR2 > basophil, which is not suitable for our single cell-based CCI identification.

We found that thousands of TCGA cancer samples have genetic mutations in the 84 genes. By comparing those with samples without any mutations in the 84 cytokine and receptors, we explored patient survival, mutational burden, and cancer stage information. Since these observations are based on the TCGA pan-cancer datasets, tumor mutation burden refers to the total number of alterations found within the TCGA tumor genomes. The presence of high TMB levels increases the likelihood of neoantigens being generated, which distinguishes cancer cells from normal, healthy cells (*Jhunjhunwala, Hammer & Delamarre, 2021*). It is possible for the immune system to identify cancer cells as foreign invaders since high levels of mutational load are believed to enhance antigen presentation to T cells. This, in turn, increases the likelihood of tumors being identified by broadening the repertoire of T cells that can kill tumor cells (*Mpakali & Stratikos, 2021*).

## CONCLUSIONS

Our data provide the first evidence that cytokines and their receptors, when mutated or actively expressed in tandem with one another, may be associated with increased mutational burden and promotion of the immune response. In many scenarios, these single cell-based communications may influence the tumor microenvironment; therefore, these cytokine- and receptor-based biomarkers may be translated into prognostic predictors for cancer immunology.

### Funding

This work was supported by the National Key Research and Development Program of China (No. 2017YFC1201200) and the National Natural Science Foundation of China (grant no. 31671375 and 31871339). The funders had no role in study design, data collection and analysis, decision to publish, or preparation of the manuscript.

### Grant Disclosures

The following grant information was disclosed by the authors:
National Key Research and Development Program of China: 2017YFC1201200.
National Natural Science Foundation of China: 31671375, 31871339.

### Competing Interests

Min Zhao is an Academic Editor for PeerJ.

### Author Contributions

- Yining Liu performed the experiments, analyzed the data, prepared figures and/or tables, authored or reviewed drafts of the article, and approved the final draft.
- Min Zhao conceived and designed the experiments, performed the experiments, authored or reviewed drafts of the article, and approved the final draft.
- Hong Qu conceived and designed the experiments, authored or reviewed drafts of the article, and approved the final draft.

### Data Availability

All data is available at the UCSC genome browser: Available at https://genome.ucsc.edu/cgi-bin/hgTrackUi?db=hg38&position=chr1%3A11102837-11267747&g=gdcCancer.

### Supplemental Information

Supplemental information for this article can be found online at http://dx.doi.org/10.7717/peerj.16221#supplemental-information.

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
