# Peer review of "Identification of cytokine-induced cell communications by pan-cancer meta-analysis"

_PeerJ, doi:10.7717/peerj.16221_

## Round 0.1 · original submission · Major Revisions

Thank you for submitting your work to PeerJ. We have received the comments from the reviewers. Two of the reviewers found the paper to be highly informative, but they have raised several queries that require clarification. One reviewer recommended rejecting the article, citing various missing aspects. The primary concern highlighted by this reviewer is the lack of sufficient information in the methods and results section, which hampers reproducibility. In light of the feedback from all three reviewers, we would like to offer the authors an opportunity to revise the manuscript, addressing the raised queries.

Reviewer 1 ·

Basic reporting

In my opinion, the article is lacking proper structure, with missing information in introduction, excessive information in methods and results which would be more appropriate from introduction or results and insufficient information in methods and results that is relevant to reproducing the authors work and describing the results. I have included specific comments below -

1. In the Introduction, sufficient literature has not been cited. In the uploaded docx file, I have highlighted all the places where proper citation must be included.

2. I believe the introduction should briefly explain what transient CCIs are from a molecular biology perspective.

3. Methods lines 106-110
This is general information about scRNAseq and not relevant to the ‘methods’ needed to reproduce authors study.

4. Methods lines 123-124,
This sentence may be more appropriate for the results.

5. In Results, lines 142 - 148
This paragraph can be moved to Introduction/Discussion or can be summarized in a single sentence for the results.

6. In results, lines 150-158,
This paragraph reads more like a methods section. There is little ‘Results’ information in it (with the exception of 155-156)

7. In Results lines 162, please specify the dataset and cite it.

8. Please clarify the following in lines 282-288 -

8.1 "In general, more genetic changes may result in the presentation of a greater number of antigens at the cell surface, increasing the chance of an immune response being initiated"
This would occur only in specific types of variants such as CNVs, or mutations directly or indirectly leading to increased expression of cytokines.
8.2 "This also explains why a high mutational burden in less common solid tumors is often associated with patient survival [20]."
Are the authors referring to poorer survival?
8.3 "On the contrary, mutated samples are more likely...."
Specify which samples

Experimental design

In my opinion, the article is missing key information necessary to reproduce the authors' work by a readers, particularly in the methods. I have included specific comments below -

8. Methods, lines 84-98
I am not certain if this is relevant for methods and may be more appropriate for Introduction or discussion. Please specify the parameters that were input in cancerSEA to shortlist and download the data.

9. lines 100-104
Please be specific about the parameters used to shortlist the total pairs of cytokines.

10. lines 105-112
Did the authors align the raw data or only work with read counts? How did the authors determine cells as ‘empties’ or ‘dropouts? What platforms were used to generate the various scRNAseq data and how did authors normalize the data for different platforms, different cell types?

11. lines 126-130
Methods should include specifics of the analysis conducted. Authors should include versions of R and R package. Did authors use R Studio. What statistical tests were included for pathway analysis? Was there a cut-off for number of genes per pathway or number of cytokine-ligand pairs per pathway?

12. In lines 254-265
Please include the name of the statistical tests used, and p-value correction methods etc, in the parenthesis.
Please also specify which analysis this was - IPA enriched pathways, gene ontology (GO) biological processes

13. In Results, lines 294-308
A good first step would be to combine data from the many available datasets mentioned earlier in the manuscript and plot them in a single t-SNE or UMAP showing similar cell types across different datasets showing a similar type of cytokine expression profile. After this, additional t-SNEs or UMAPs could be plotted with the cell type and pathways data overlapped.

Validity of the findings

Without sufficient detail in the methods, it is difficult to reproduce the authors work and therefore difficult to assess the validity of the findings.

I am also not confident that authors have provided sufficient evidence to support the claims made in conclusions, particularly those in lines 329-334.

Additional comments

1. The abstract is well written. Please include a brief one-sentence summary of dysregulated cell types or cell-type unique to tumors.

2. In line 144, 'challenging' would be more appropriate in place of 'impossible'

Annotated reviews are not available for download in order to protect the identity of reviewers who chose to remain anonymous.

Reviewer 2 ·

Basic reporting

No comment

Experimental design

No comment

Validity of the findings

No comment

Additional comments

in this manuscript, the authors performed the pan-cancer meta-analysis of cytokine and receptor interactions at the single-cell level. They found that the cytokines-receptor binding of interleukins and growth factors regulates the tumor microenvironment, which makes these cytokines and their related receptor promising targets for cancer therapy.

The article is overall well-written and scientifically profound.

·

Basic reporting

Clear and professional English is used throughout.
Figure fonts should be changed to enhance readability. I can’t read the text of Figures 2 and 3, even at 200% zoom.

Experimental design

No comment

Validity of the findings

No comment

Additional comments

In the manuscript, the author has done a pan-cancer meta-analysis to identify cytokine-induced cell communication. Overall, in my opinion, the idea is interesting and holds promise for advancing our understanding of the complex interplay between cytokines and cancer biology. The overall writing seems fine. But I have some concerns:
1) The statistic available on the home page of CancerSEA (http://biocc.hrbmu.edu.cn/CancerSEA/) clearly mention the number of cancer types as 27 and the number of cancer single cells as 93475. I assume the possibility that this number may be increased at this time and when authors have downloaded the data, it might be 25 and 41900. To clear this discrepancy in the number, the date of data download must be included in the method as well as the cancer type. So the reader can get which cancer type has not been included in your study.
2) The equation of cell communication ratio is interesting. If the equation is already established, then proper referencing should be done. Although, as per my limited knowledge, I didn’t come across with any such method of defining cell communication. Armingol et al wrote a comprehensive review about deciphering cell-cell interaction and communication in “Nature review genetics” and I don’t come across any method of calculating the cell communication ratio. Could you please provide further justification for the proposed equation? It would be helpful to understand the underlying principles, assumptions, and empirical evidence that support its validity. Additionally, references to relevant literature or previous studies would enhance the credibility and clarity of the equation.
3) There are available methods such as CellChat for the analysis of cell communication using scRNA-seq data. The authors completely ignore discussing these methods.
4) Besides, there are methods for cytokine communication mapping (although based on RNAseq). What is the limitation of these methods? A short comparison or discussion about the principle, utility, pros, and cons will be appreciated.
5) The figure font should be changed to enhance readability. I can’t read the text of Figures 2 and 3, even at 200% zoom.

---

## Round 0.2 · accepted · Accept

The authors have incorporated most of the comments and suggestions by both reviewers. While I am happy to accept this manuscript for publication, there are a few suggestions by one of the reviewers that need to be considered.

Reviewer 1 ·

Basic reporting

Authors have addressed most of the comments on basic reporting, added additional literature refrences and improved figures for readability. I only have one further comments to make

- Reponse to Basic Reporting comment 3

While describing "dropouts", please mention "lowly" expressed transcript, rather than just 'expressed transcript'

Experimental design

Authors have satisfactorily addressed most of my comments. I have reviewed the cahnges the authors have made as well as their rebuttal, and have the following comments/suggestion:

- Response to comment 8
As a supplemental, please include the R code used to generate these results.

- Response to comment 10
Please ensure that these details (regarding TPm expression values) are included and mentioned in your methods, and original literature cited.

- Response to comments 13
A combined analysis could still be useful to identify or discover pathogenic cell types with commonalities across cancer types.
These could be angiogenesis related endothelial type cells, altered fibroblasts or immune cells, with non-functional CCIs.
Identifying these CCIs could be useful for target identification for cancer therapeutics. Could cell-based therapies aimed at restoring these CCIs be developed?

Validity of the findings

no comment

Additional comments

Please briefly mention the grouping criteria for CellChat, are they using the groups of similar types of cells, or cell originating from similar tissue, etc.
Please also clarify, if and how your method (as well as other methods) accounts for excluding cell-types, and their corresponding cytokines that are unlikely to interact (based on spatial or other constraints).
Would data driven constraints be better for setting cutoffs?

·

Basic reporting

Everything is up to the place from English writing to literature reference to figure quality.

Experimental design

Experimental setup is fine and justified with the writing.

Validity of the findings

This manuscript will surely impact the field of cell-cell communication.